# Caveolin-1: A Review of Intracellular Functions, Tissue-Specific Roles, and Epithelial Tight Junction Regulation

**DOI:** 10.3390/biology12111402

**Published:** 2023-11-05

**Authors:** Cody M. Dalton, Camille Schlegel, Catherine J. Hunter

**Affiliations:** 1Division of Pediatric Surgery, Oklahoma Children’s Hospital, 1200 Everett Drive, ET NP 2320, Oklahoma City, OK 73104, USA; camille-schlegel@ouhsc.edu (C.S.); catherine-hunter@ouhsc.edu (C.J.H.); 2Health Sciences Center, Department of Surgery, University of Oklahoma, 800 Research Parkway, Suite 449, Oklahoma City, OK 73104, USA

**Keywords:** Caveolin-1, caveolae, tight junction, barrier function, cellular processes

## Abstract

**Simple Summary:**

Caveolin-1 (Cav1) is a protein that exists in many different forms and locations in cells and tissues throughout the body. We can understand more about cell growth, death, and cellular processes by further understanding the structure and function of Cav1. The increasing knowledge of Cav1 and its roles in different organs and disease processes helps delineate its potential use in the development of treatments and therapies.

**Abstract:**

Caveolin-1 (Cav1) is a vital protein for many cellular processes and is involved in both the positive and negative regulation of these processes. Cav1 exists in multiple cellular compartments depending on its role. Of particular interest is its contribution to the formation of plasma membrane invaginations called caveolae and its involvement in cytoskeletal interactions, endocytosis, and cholesterol trafficking. Cav1 participates in stem cell differentiation as well as proliferation and cell death pathways, which is implicated in tumor growth and metastasis. Additionally, Cav1 has tissue-specific functions that are adapted to the requirements of the cells within those tissues. Its role has been described in adipose, lung, pancreatic, and vascular tissue and in epithelial barrier maintenance. In both the intestinal and the blood brain barriers, Cav1 has significant interactions with junctional complexes that manage barrier integrity. Tight junctions have a close relationship with Cav1 and this relationship affects both their level of expression and their location within the cell. The ubiquitous nature of Cav1 both within the cell and within specific tissues is what makes the protein important for ongoing research as it can assist in further understanding pathophysiologic processes and can potentially be a target for therapies.

## 1. Introduction

Caveolin-1 is a highly ubiquitous protein found in many tissues throughout the human body. It is involved in processes including cell membrane maintenance, cellular signaling, differentiation, proliferation, and migration as well as regulation of programmed cell death and autophagy [1,2,3,4,5,6,7,8,9,10,11,12,13,14,15,16,17,18,19,20,21,22,23]. This diverse set of roles varies depending on the tissue-specific location of caveolin-1 (Cav1), which includes but is not limited to endothelium, adipose, pneumocytes, and muscle and intestinal epithelium [24,25,26,27,28,29,30,31,32,33,34,35,36,37,38,39,40,41,42,43,44,45,46]. It is recognized that Cav1 is implicated in both the positive and negative regulation of many cellular processes [47,48,49,50,51,52]. For example, downregulation of Cav1 in the early stages of tumorigenesis of certain cancer types promotes proliferation, angiogenesis, and tumor progression while in the later stages re-expression of the protein seems to support cell invasion and metastasis [53]. This seemingly paradoxical description of function is one of many reasons that Cav1 is an important and interesting target protein for ongoing research and discovery. Cav1 has been studied as a critical participant in cytoskeletal interactions, intracellular signaling, cholesterol transport, and tight junction regulation (Figure 1) [38,39,40,46,47,48,49,50,51,52,54,55,56,57,58,59,60,61,62,63,64,65,66,67,68,69,70,71,72,73,74,75,76,77]. The latter is especially important in the maintenance of the epithelial barrier. The synthesis and proper placement of tight junctions within the plasma membrane of epithelial cells can be altered in many epithelial barrier diseases. This review will focus on the structure and overall function of Cav1 and how it is utilized in different tissue types for fundamental cellular mechanisms. There will be a concentration on its specific actions within intestinal epithelial cells and how these actions are altered in disease. 

## 2. Caveolin-1 Structure and Function

Cav1 is a 21 kDa protein that is associated with the plasma membrane having two hydrophilic termini residing on the cytosolic side separated by a hydrophobic segment lying within the membrane [78]. Here it performs its many functions with its two main functional domains: the tyrosine 14 phosphorylation site and its scaffolding domain [78]. The scaffolding domain is responsible for interactions with other proteins that regulate intracellular signaling cascades [78]. The Src family proteins, G proteins, protein kinase A and C, tyrosine kinase receptors (TKRs), and nitric oxide synthase (NOS) are just a few of these regulator proteins [79,80,81,82]. Phosphorylation of Cav1 on specific amino acid residues causes upregulation or downregulation of certain processes such as cholesterol trafficking [83,84]. Cav1 is also involved in the formation of caveolae. Caveolae, first described in 1955 while looking at gallbladder epithelium in mice, are invaginations of the cell plasma membrane that form vesicles which communicate extracellularly [85]. Interestingly, Cav1 is a protein that is palmitoylated on three cysteine residues on the C-terminus [86,87]. This stabilizes the protein–membrane relationship and is required for certain protein–protein interactions as is the case with G proteins [86,87]. Cav1 has been demonstrated to induce cholesterol clustering and higher degrees of membrane curvature in cholesterol-rich membranes [86]. This is a process critical for cell signaling and the formation of caveolae [86]. Cav1 was first thought to perform only absorptive functions via caveolae formation without any mention of its secretory or intracellular functions [85]. However, we now have substantially more knowledge regarding the structural complexity of Cav1 and the variety of functions it participates in. 

Ohi et al. improved the understanding of the intricate structure of Cav1 by describing its process of oligomerization prior to caveolae assembly [88]. The disc-like 8S oligomeric complexes go on to organize as subunits of a more sophisticated 70S structure that is shuttled to the plasma membrane where it performs many functions such as caveolae formation and interactions with the cytoskeleton [89]. Whereas it was previously thought that caveolae were a continuously curved shape, the most recent model suggests a polyhedral shape with the 8S complexes of Cav1 forming the flat surfaces of each polygonal surface [90,91,92,93,94]. Through its interactions with the actin cytoskeleton and formation of caveolae, Cav1 assists in cell cycle regulation, endocytosis, cholesterol trafficking, and efflux [54]. Coordination with the cellular cytoskeleton allows for Cav1 to play a role in mechanosensitization and plasma membrane restructuring in response to extracellular stress [54]. Cav1 has tissue-specific functions and there is evidence that it is extensively involved in processes that determine cell proliferation, differentiation, and survival [1,2,3,4,5,6,7,8,9,10,11,12,13,14,15,16,17,18,19,20,21,22,23,47,48,49,50,51,52]. Knowing the structure and function of Cav1 allows for a higher understanding of how the protein might contribute to specific disease states, including those that effect the intestinal epithelial barrier.

## 3. Cellular Signaling

Cav1 serves as a key component of multiple cellular signaling pathways and both positive and negative regulatory functions have been identified [47,48,49,50,51,52]. For example, there is evidence of Cav1 involvement in the p38 mitogen-activated protein kinase (MAPK) signaling pathway, by which decreased Cav1-mediated activation of c-Jun N-terminal kinase (JNK), nuclear factor kappa B (NF-kB), and activator protein-1 (AP-1) leads to protection against inflammatory mechanisms in the setting of sepsis [47]. In contrast, Cav1 also interacts with toll-like receptor 4 (TLR4), a protein that activates the innate immune system, and increases endothelial cell permeability in the setting of hypoxia [48]. When Cav1 is knocked out, the inflammatory response mediated through the TLR4 pathway is suppressed [49]. Cav1 also has a role in modulating cellular responses to oxidative stress and inflammation by regulating the production of reactive oxygen species via NADPH oxidase activation and downstream signal transduction [50,51,52]. The versatile nature of Cav1 in regulating cellular processes is integral to understanding its role in cell growth, migration, death, and survival.

## 4. Cell Proliferation, Differentiation, Migration and Survival

### 4.1. Proliferation

Cav1 plays a part in multiple mechanisms that maintain cell viability and stabilization [1,2,3]. In general, Cav1 is noted to have inhibitory effects on cell proliferation as demonstrated in both mouse and human Cav1 knockout mesenchymal stem cells in which proliferation increased. There is further evidence of this phenomenon with multiple studies showing that upregulation of Cav1 has the opposite result and decreases proliferation [1,2]. In vivo, Cav1 overexpression arrested the cell cycle in mouse embryonic fibroblasts at the G0/G1 phase causing growth and mitotic activity to come to a halt [3]. Furthermore, the cells took on a senescence-like morphology and increased B-galactosidase activity which is associated with senescence [3].

### 4.2. Differentiation

There is evidence to suggest that Cav1 has both inhibitory and promoter effects on cellular differentiation [50,51]. Knocking out Cav1 in mouse stem cells decreased mRNA expression of pluripotency markers while overexpression of Cav1 had the opposite effect [4]. It is hypothesized that Cav1 activity may allow progenitor cells to be more sensitive to certain stimuli that cause ongoing differentiation [4]. As cells reach their final phenotype, Cav1 can then function as part of a negative feedback loop to prevent ongoing growth and differentiation [5]. This was also demonstrated with mesenchymal stem cells undergoing osteogenic differentiation as Cav1 activity enhanced the process before stabilizing the final phenotype [6,7]. Conversely, as an example of Cav1 having a primarily inhibitory role, prolactin suppresses Cav1 expression in mammary epithelium in pregnant mice [7]. This indicates that further growth and differentiation of those cells during pregnancy and lactation requires Cav1 to be downregulated [7].

### 4.3. Migration

One illustration of how Cav1 influences cell polarity and migration is how the protein assists with stromal-derived factor-1 (SDF-1) interactions with its receptor, CXCR4. This plays a crucial role cell adhesion during leukocyte recruitment [8]. Cav1 is required for mobilization of progenitor cells from mouse bone marrow [9]. Through interactions with the cytoskeleton, Cav1 promotes cellular polarity and morphological changes that allow for cellular migration [9]. This was further found to be regulated via Cav1 activation of Src and Rho GTPase [10]. Grande-Garcia et al. found that Cav1-deficient mouse embryonic fibroblasts did not exhibit polarized morphology via actin rearrangements [10]. Cells can also relocalize Cav1 depending on the type of movement they are performing [10,11]. During three-dimensional movement, endothelial cells release Cav1 from caveolar structures in the cell rear and relocalize it to the front [11]. Contrastingly, in planar movement Cav1 remains in the rear to colocalize with caveolae [11]. Through its activity in cellular movement, adhesion, and interactions with extracellular receptors, Cav1 is important for processes such as inflammation, wound healing, and angiogenesis [11].

### 4.4. Survival

Cav1 involvement in the regulation of cell survival and programmed death in response to stress or insult has important implications for tissue repair and wound healing. For example, downregulation of Cav1 expression in myogenic precursor cells is mandatory for regeneration and healing in skeletal muscle tissue [12]. Dysregulation of these mechanisms is pertinent to certain disease states. Cav1 is an integral player in both cell survival and apoptosis. Apoptosis is cell death that can occur as a controlled part of an organism’s natural growth cycle or in response to extra or intracellular stressors. Just as essential to the cell cycle is the process of autophagy, in which Cav1 has also been shown to play a major role. Autophagy is the degradation of damaged or unnecessary cellular components through a lysosome-dependent mechanism. This process can be part of an adaptive response that promotes cell survival in the setting of starvation. There are multiple examples of Cav1 being a positive regular of autophagy and a negative regulator of apoptosis [13,14,15]. It was concluded by Nah et al. that Cav1 activated autophagy in oxidative stress and cerebral ischemic injury by localizing the BECN1 complex to mitochondria in order to protect against ischemic [13]. This did not happen in the absence of Cav1 and cerebral ischemic damage worsened. In human breast cancer cells, Cav1 knockout resulted in decreased expression of autophagy-related proteins (LC3-II and Atg12/5), autophagosome formation was inhibited, and apoptosis ensued [14]. Aflatoxin, a known hepatotoxic substance, has a dampened ability to induce apoptosis in the absence of Cav1, while also inducing autophagy via regulation of the EGFR/PI3-AKT/mTOR signaling cascade [15]. The specific manner in which Cav1 contributes to apoptosis has been suggested by multiple studies, one of which describes a process independent from the activation of the well-known caspase-induced DNA fragmentation process [16]. It was observed that Cav1 also correlates with increased levels of phosphatidylserine expression at the cell surface, which is a marker of cells undergoing the death process [16].

### 4.5. Caveolin-1 Role in Cancer Cell Death and Survival

The role of Cav1 in tumor growth, metastasis, and response to treatment appears to be most closely linked with its role in regulating apoptosis and cell death mechanisms. Cav1 can act in accordance with tumor suppressor or promotor modalities depending on the cell type, tumor stage, and nature of the tumor stroma [17,18,53]. In specific cancers including breast, lung, colon, and ovarian, Cav1 is expressed in low levels, which is maintained throughout the process of tumor cell proliferation and metastasis [19,20,21,22,23]. Contrastingly, Cav1 is downregulated in oncogenically transformed cells, but then upregulated again in later tumor stages to potentially support tumor invasion and drug resistance, which has been demonstrated in prostate cancer [19,20,21,22,23]. Again, this reveals the complexities of the Cav1 protein and its functions. Its ability to perform opposing oncologic functions depending on cell type or stage of disease exposes the importance for determining these nuances within other disease processes.

## 5. Caveolin-1 Tissue-Specific Roles

To further understand the roles that Cav1 plays in individual tissue types, it must first be explained that Cav1 takes on multiple different forms within the cell itself in order to perform these specific functions. As a membrane protein, Cav1 forms caveolae to interact with cytoskeletal elements and signaling cascade proteins [47,48,49,50,51,52,54]. In the cytoplasm, it is a soluble protein that participates in transporting and delivering lipids to the cell surface, mitochondria, endoplasmic reticulum, and other cellular compartments [24]. Examples of cell types in which Cav1 is localized to the membrane in the form of caveolae include fibroblasts, endothelial, and polarized epithelial cells of the intestine [24]. However, for instance, in pancreatic acinar cells, Cav1 exists mainly in cytosolic and mitochondrial forms performing in secretory pathways [24]. In some cells, Cav1 is in multiple compartments simultaneously and can move in between them based on certain cellular stimuli. [25,26]. In this way, Cav1 location and function is tailored to the needs of the cell depending on the specific tissue in which it resides.

### 5.1. Vascular

Cav1 expression in vascular endothelial cells regulates the progression of atherosclerosis in large vessels [27]. Increased autophagy in Cav1 knockout cells has shown to protect against the progression of atherosclerosis via a decrease in vascular inflammation, macrophage infiltration, and LDL transcytosis [27]. Cav1 also inhibits endothelial nitric oxide synthase activity via direct binding of endothelial nitric oxide synthase (eNOS) and regulation of its expression [28]. Reciprocally, eNOS-derived NO causes Cav1 ubiquitination and degradation in a negative feedback pathway [28]. This is evidence that Cav1 helps to maintain vascular homeostasis [28]. Cav1-mediated endothelial cell migration via activation of VEGF receptors within caveolae is also crucial for NO-mediated angiogenesis [29]. Cav1 participation in vascular function and smooth muscle contraction is further demonstrated by the development of pulmonary hypertension and right-ventricular hypertrophy seen in Cav1 knockout mice [30]. This normalized with Cav1 reconstitution in the pulmonary vascular endothelial tissue [30].

### 5.2. Adipose

Adipose tissue has been described as one of the most abundant sources of caveolin proteins [31]. Caveolins are established cholesterol binding proteins [31,78]. Exogenous, extracellular lipids induce plasma membrane caveolins to associate with lipid droplets in adipocytes [32,33]. This indicates a trafficking pathway by which Cav1 shuttles cholesterol and fatty acids from plasma membrane to the organelle and vice versa [32,33]. In Cav1 null mice there is evidence of aberrant lipid metabolism, namely a lipoatrophic phenotype and hyperlipidemia, which results from reduced lipid storage within adipocytes and therefore increased levels of lipids remaining in circulation [34]. Cav1 null mice also have resistance to diet-induced obesity [35]. Cav1 performs key interactions with insulin receptors by affecting downstream signaling and stabilizing the receptors to the plasma membrane [32]. During states of insulin resistance, dissociation of the insulin receptor from Cav1 results in defective downstream insulin signaling, GLUT4 translocation to the plasma membrane, and thus glucose uptake by the cell [36,37]. This suggests that inhibiting this process could be a therapeutic target for T2DM and proposes the importance of ongoing exploration of Cav1 and its functioning within metabolic processes [36,37].

### 5.3. Brain

There are a host of factors that influence the integrity and permeability of the blood-brain barrier (BBB). Caveolins are thought to perform vital functions in the regulation of the junctional proteins that create the barrier between vascular endothelial cells and the brain parenchyma [38,39,40]. Choi et al. delineated this relationship by showing that Cav1 null mice had significant degradation of tight junction proteins and an increase in matrix metalloproteinase (MMP) proteolytic activity after exposure to ischemia when compared to wild-type mice [38]. Interestingly, Cav1 knockout also showed impaired angiogenesis and increased apoptosis at the BBB after ischemic insult [39]. There is substantial evidence that the loss of BBB integrity in the setting of ischemia and the resulting cerebral edema is closely related to tight junction dysregulation, a mechanism that will be further discussed as it pertains to other biological barrier functions [40].

Caveolin-1 has also been shown to organize synaptic signaling molecules within membrane-lipid rafts, which has important implications for Alzheimer’s Disease [41]. There is evidence that this relationship also plays a preventative role in amyloid precursor protein processing and amyloid-β toxicity [41]. In the setting of Cav1 overexpression, amyloid precursor protein and β-secretase localization to membrane-lipid rafts resulted in decreased Aβ production [41]. Conversely, knocking out Cav1 resulted in significantly reduced synapses, while re-expression of Cav1 decreased Aβ expression [41]. Additionally, Cav1 expression increases in the brain with age [42]. This leads to increased intercellular transmission of α-synulcein which leads to the progression of Parkinson’s Disease [42]. This process was attenuated in the setting of Cav1 inhibition [42].

### 5.4. Pneumocytes

Caveolin-1 is essential for pulmonary function and development. As previously discussed, Cav1 is involved in the proliferation and differentiation of stem cells [1,2,3,4,5,6,7]. This process is also true for fetal lung development, specifically as it relates to the differentiation into type II alveolar epithelial cells, also known as pneumocyte [43]. Type II pneumocytes have a high expression of Cav1 and structurally contain an abundance of caveolae [44,45]. It is thought that these caveolae are partly responsible for maintaining the homeostasis of water and protein transcytosis across the epithelial membrane [43,44,45,46]. In bronchopulmonary dysplasia, exposure of type II pneumocytes to hyperoxia showed disruption of the pulmonary epithelial barrier [46]. These cells had decreased mRNA expression and protein levels of the tight junction proteins zona occludin-1, occludin, and claudin-4 as well as significantly decreased colocalization with Cav1 in the cell membrane [46]. This was also seen in Cav1 knockout mice even under normoxia conditions and was reversed when Cav1 was upregulated [46]. Cav1 thus has a close relationship with epithelial barrier integrity and permeability.

## 6. Caveolin-1 and the Intestinal Barrier

Multiple components contribute to the maintenance of the intestinal barrier. The intestinal epithelial layer acts as a semipermeable membrane allowing the absorption of nutrients and performance of immune functions while preventing the transport of harmful substances [55]. The mucosal layer is the first line of defense for epithelial cells coming into direct contact with potentially toxic microorganisms. IgA and antimicrobial proteins are secreted into this layer in order to neutralize these microorganisms [56]. The layer of polarized epithelial cells that separates the intestinal lumen from the lamina propria underneath is highly selective in what is allowed to be transported across [55]. This is in part due to regulation via the presence of junctional complexes, which consist of tight junctions, adherens junctions, and desmosomes [57,58]. Tight junctions determine a healthy epithelial barrier. Disruption of the tight junctions by systemic stressors can lead to intestinal hyperpermeability and trigger systemic immune responses [59]. These components of the intestinal barrier are the major points of interest when researching and describing the pathogenesis of gastrointestinal diseases.

Although Cav1 is seen in the cytosolic and intracellular compartment forms, it is most commonly thought to be associated with the cell surface [24]. One purpose this might serve is to integrate junctional complexes into the plasma membrane. This anchors epithelial cells at the basolateral surface both to each other and to the basement membrane (Figure 2). The location of junctional complexes within the cell and their level of expression is, in part, Cav1-dependent [60,61,62,63,64,65,66,67]. Cav1-mediated shuttling and endocytosis of tight junctions during their relocation has been described previously [60,61,62,63,64,65,66,67]. Marchiando et al. showed specifically that occludin sequestration was the first morphological change and that the mechanism by which occludin was relocated was Cav1-dependent as both proteins were found to be colocalized [68]. Additionally, with the utilization of Cav1 knockout mice, they demonstrated that after immune activation with tumor necrosis factor alpha (TNF-a), enterocytes failed to internalize occludin [68]. This holds true across multiple studies that have demonstrated the relationship of Cav1 with tight junctions like occludin. Leiva et al. performed in vivo experiments using human intestinal tissue collected from infants at time of exploratory laparotomy for patients either with or without necrotizing enterocolitis [69]. Using immunofluorescence and western blot analysis it was shown that both Cav1 and occludin were decreased in concentration within enterocytes, also suggesting a relationship between Cav1 and protein expression of tight junctions like occludin [69]. Other studies have revealed this phenomenon and provided further evidence for Cav1 involvement in endothelial cell permeability and barrier function. For example, inducing actin depolymerization leads to caveolae-dependent endocytosis of tight junction components and knocking out Cav1 results in the loss of thrombin-induced opening of cell junctions by weakened associations with cadherins [65,70].

Additionally, claudins are transmembrane proteins that also function as tight junctions. Claudin-2 is known to be a pore-forming protein that allows for the passage of extracellular contents in paracellular transport in the intestinal barrier [71,72,73,74]. Ares et al. found that claudin-2 is overexpressed in the setting of inflammatory stressors and that there is increased binding with Cav1 [71]. The relationship between Cav1 and claudin-2 as well as occludin is evident. However, other tight junctions such as claudin-4 and zona occludin do not seem to interact with Cav1 and likely have actions that are independent of the protein [75]. Cav1 and its important interactions with tight junctions like occludin and claudin-2 is just one of many factors required for maintenance of the epithelial intestinal barrier. Changes in Cav1 expression and localization has downstream effects that can lead to increased intestinal permeability.

## 7. Conclusions

Caveolin-1 is a vital protein for many cellular processes and its involvement in both the positive and negative regulation of these processes has been clearly demonstrated. Its pervasiveness throughout a variety of tissue and cell types has established the protein as a point of interest for many researchers to better understand the pathophysiologic mechanisms of disease. Cav1 as a plasma membrane protein carries out regulatory functions of many intracellular signaling cascades. Here it also creates invaginations in the plasma membrane called caveolae and interacts closely with cytoskeletal structures. This allows the protein to assist in critical cellular functions such as cell cycle regulation, endocytosis, cholesterol trafficking, and efflux. Cav1 is implicated in stem cell differentiation and it acts in a negative feedback loop to prevent further differentiation as cells reach their final phenotype. This function in addition to its negative regulation of cellular proliferation is of particular interest when studying cancer and aberrant cellular processes that allow for unchecked cell growth and division. Additionally, Cav1 generally acts as a positive regulator of cell autophagy and a negative regulator of apoptosis. In tissues that have experienced damage, such as in the setting of cerebral ischemic injury, this can allow for protection against ongoing ischemic change. Cav1 has tissue-specific functions that are adapted to the requirements of the cells within those tissues. Its role has been described in adipose, lung, pancreatic, and vascular tissue, to name a few. Cav1 is also involved in epithelial barrier maintenance. In both the intestinal and the blood brain barriers, Cav1 has significant interactions with junctional complexes that manage barrier integrity. Tight junctions have a close relationship with Cav1 and this relationship affects both their level of expression and their location within the cell. In intestinal tissue for example, in the presence of external cellular stress factors, occludin is internalized from the plasma membrane in a Cav1-dependent process and both proteins are found to be colocalized within the cell. Cav1 and its relationship with tight junctions is a current point of interest in pathologic processes where barrier breakdown causes systemic disease.

## 8. Future Directions

Caveolin-1 requires further investigation as it pertains to a multitude of physiologic and pathologic processes. Its ubiquitous nature makes it a challenging protein to study as it is pervasive throughout many organ systems, tissue types, and pathophysiologic processes. The advent of transgenic and gene knockout technology continues to allow for the determination of cellular effects in both the presence and absence of the protein [95]. However, continued work must be conducted to establish the conditions and mechanisms by which Cav1 expression and behavior may be affected. For example, we know that Cav1 takes part in signaling cascades consistent with both tumor suppression and promotion, however, understanding how and when one set of characteristics is expressed over the other remains to be completely explained [76,96]. Additionally, some cellular processes in which a role for Cav1 has been demonstrated can still be carried out when Cav1 is deleted [97]. This raises the question of what backup mechanisms exist to maintain biological redundancy when or if the Cav1 system fails [77,97]. As our understanding of Cav1 and its contribution to these mechanisms increases we will hopefully be able to utilize it as a biological marker or predictor of disease, and ultimately as a potential therapeutic target for a multitude of disease processes.

## Figures and Tables

**Figure 1 biology-12-01402-f001:**
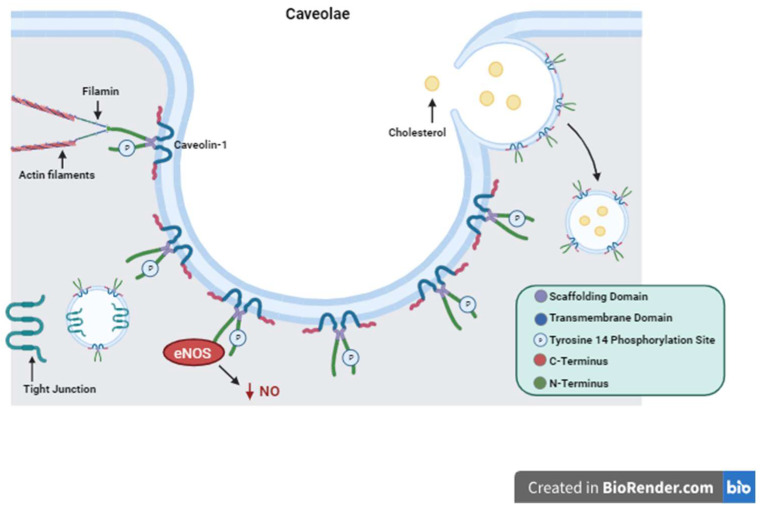
Caveolin-1 Structure and Function. Depiction of Caveolin-1 structure including its scaffolding domain, transmembrane domain, and Tyrosine 14 phosphorylation site. The structure of caveolae is represented with Caveolin-1 proteins forming the surface of the plasma membrane invagination. Important functions of Caveolin-1 are illustrated showing cytosolic interactions with actin filaments, endocytosis and cholesterol trafficking, tight junction shuttling, and intracellular signaling eNOS inactivation and decreased production of NO as an example.

**Figure 2 biology-12-01402-f002:**
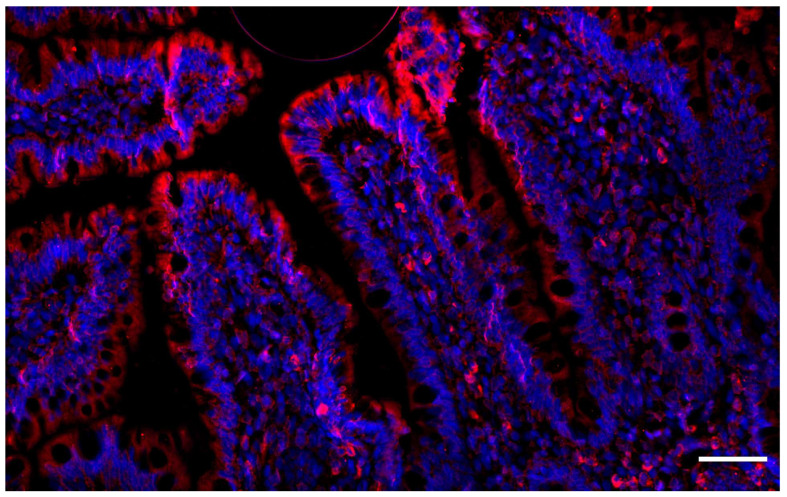
Caveolin-1 Localization. Immunofluorescence staining of healthy small intestinal tissue showing a cross-sectional image of intestinal villi and polarized epithelial cells. Caveolin-1 (red) is seen localized to the apical and basolateral membranes rather than cytosolic compartments. Image taken at 20× magnification, the scale bar on the bottom right indicates 50 μm. Epithelial cell nuclei are stained blue using Fluroshield with DAPI (Sigma, St. Louis, MO, USA, CAT#F6057). Caveolin-1 is stained red using primary antibody (Novus, Centennial, CO, USA, CAT#NB100-615) and secondary antibody (Invitrogen, Waltham, MA, USA, CAT#A11032).

## Data Availability

Data sharing not applicable.

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
