# Peer review of "Caveolin-1: A Review of Intracellular Functions, Tissue-Specific Roles, and Epithelial Tight Junction Regulation"

_biology, 2023, doi:10.3390/biology12111402_

Round 1

Reviewer 1 Report

Comments and Suggestions for Authors

The manuscript titled "Caveolin-1: A Review of Structure, Function, and Tissue-specific Roles" provides a succinct and concise summary of caveolin-1 functions which can be found most useful to the readers of the journal and scientists working with this mulifaceted molecule.  Authors must review some statements to insert the corresponding references on the functional roles of caveolin-1.  Further, the title seems overambitious in its dimensions particularly on the structural aspects of caveolin-1, considering recent and classical considerable studies on its structure (i.e. Ohi MD, Kenworthy AK. Emerging Insights into the Molecular Architecture of Caveolin-1. J Membr Biol. 2022;255(4-5):375-383. doi:10.1007/s00232-022-00259-5). Hence a more appropriate title can be chosen which builds upon the manuscript's strength and main contributions of the authors to the field related to the junctional roles of caveolin -1.  This would significanctly highlight the authors' key contributions and attract and/or direct readers to an important niche in the field which the authors have excelled in. Provide these minor changes are addressed I would recommend its publication in Biology.

Author Response

Dear Reviewer,

Thank you very much for taking the time to review this manuscript. Please find the detailed responses below and the corresponding revisions/corrections highlighted.

Point 1:  Authors must review some statements to insert the corresponding references on the functional roles of caveolin-1.

Thank you for pointing this out. I agree that there were many statements throughout the manuscript that require references. Below you can see the submitted manuscript with the appropriate additions of corresponding references where they were lacking before. Changes have been highlighted in yellow.

Point 2: Hence a more appropriate title can be chosen which builds upon the manuscript's strength and main contributions of the authors to the field related to the junctional roles of caveolin -1.  This would significantly highlight the authors' key contributions and attract and/or direct readers to an important niche in the field which the authors have excelled in. 

I agree that the title does not quite fit the manuscript material. I believe it is too broad. The title has been difficult to create. I have revised it to see if this better fits the content of the manuscript. Changes highlighted in yellow.

Reviewer 2 Report

Comments and Suggestions for Authors

The review on “Caveolin-1: A Review of Structure, Function, and Tissue-Specific Roles” is written in an organized manner with sufficient detailing on the roles of Cav1 in different organs and disease processes. The review is relevant and important with respect to current status of the field.

The reviewer has only a few points, which could enhance the quality of the review.

11-      Authors can discuss the role of Palmitoylated Caveolin-1. Below mentioned articles can be considered for this PMID: 30158247 and PMID: 30579563

22-      Briefly discuss the role of caveolin-1 in neurodegenerative diseases. Below mentioned articles can be considered for this PMID: 21203469 and PMID: 34321069

33-      Authors can briefly discuss the challenges and limitations of current research on caveolin-1, and highlight the areas where further study is needed.

Author Response

Dear Reviewer,

Thank you very much for taking the time to review this manuscript. Please find the detailed responses below and the corresponding revisions/corrections highlighted.

Point 1:  Authors can discuss the role of Palmitoylated Caveolin-1.

Thank you for this suggestion. I did review the two suggested articles and utilized them for adding this material to the manuscript. I felt the topic best fit in the section on Caveolin-1 structure and function at the beginning of the manuscript. Please see changes highlighted in blue.

Point 2:   Briefly discuss the role of caveolin-1 in neurodegenerative diseases.

Thank you for this suggestion. I did review the two suggested articles and utilized them for adding this material to the manuscript. I felt the topic best fit in the section on tissue-specific roles and I have added it under the brain organ system section. Please see changes highlighted in blue.

Point 3: Authors can briefly discuss the challenges and limitations of current research on caveolin-1, and highlight the areas where further study is needed.

I agree that more information would be beneficial at the end of the manuscript to highlight particular challenges in studying this protein and to identify areas where further research ust be done. I have revised my “future directions” paragraph to include this. Please see changes highlighted in blue.